# ROBOGPT: AN INTELLIGENT AGENT OF MAKING EMBODIED LONG-TERM DECISIONS FOR DAILY INSTRUCTION TASKS

## ABSTRACT

Robotic agents must master common sense and long-term sequential decisions to solve daily tasks through natural language instruction. The developments in Large Language Models (LLMs) in natural language processing have inspired efforts to use LLMs in complex robot planning. Despite LLMs' great generalization and comprehension of instruction tasks, LLMs-generated task plans sometimes lack feasibility and correctness. To address the problem, we propose a RoboGPT agent[1] for making embodied long-term decisions for daily tasks, with two modules: 1) LLMs-based planning with re-plan to break the task into multiple sub-goals; 2) RoboSkill individually designed for sub-goals to learn better navigation and manipulation skills. The LLMs-based planning is enhanced with a new robotic dataset and re-plan, called RoboGPT. The new robotic dataset of 67k daily instruction tasks is gathered for fine-tuning the Llama model and obtaining RoboGPT. RoboGPT palnner with strong generalization can plan hundreds of daily instruction tasks. Additionally, a low-computational Re-Plan module is designed to allow plans to flexibly adapt to the environment, thereby addressing the nomenclature diversity challenge. The proposed RoboGPT agent outperforms SOTA methods on the ALFRED daily tasks. Moreover, RoboGPT palnner exceeds SOTA LLM-based planners like ChatGPT in task-planning rationality for hundreds of unseen daily tasks, and even other domain tasks, while keeping the large model's original broad application and generality.

## 1 INTRODUCTION

Embodied AI tasks, including visual navigation and robotic manipulation, have developed quickly (Brohan et al., 2023a). Future robots are expected to use navigation and manipulation to assist humans in carrying out more difficult daily tasks by following natural language instructions, e.g., '*make dinner*' or '*wash dishes*' (Brohan et al., 2023a;b). Some approaches, such as template planning (Inoue & Ohashi, 2022; Min et al., 2022) or expert-guided planning (Pashevich et al., 2021; Bhambri et al., 2023), have yielded some results when dealing with 7 types of daily instruction tasks. However, the existing agents cannot truly understand the instruction tasks: including object quantities, prefix content, and object dependencies, so still struggle with long-term decisions and nomenclature diversity when faced with daily tasks beyond the predefined types.

Large Language Models (LLMs) have made significant progress in the field of natural language processing (Touvron et al., 2023b). Due to their extensive internalized world information, LLMs have been investigated to solve complex robot planning problems (Song et al., 2023). However generic LLMs are overly broad and lack robotics expertise, hence the plans they produce are frequently impractical for robots to employ directly (Ahn et al., 2022; Lin et al., 2023; Song et al., 2023). For instance, given a task '*If you are a robot and give me a plan of : slice an apple*', the generated plan is '*Gather material', 'wash your hands', 'prepare the apple', '...*', which is not executable for robot. Even the LLM-Planner with detialed prompt (Song et al., 2023) produces some illogical planning of complex daily tasks.

---

[1]our code and dataset will be released soon

To address LLMs planning feasibility issues, this paper enhances the LLMs planning capability with expertise in the robotics domain, fine-tuning and improving the LLMs planning process to ensure logical validity and optimal execution. We create specialized robotic dataset of 67k robot commands includes complicated robotics activities to overcome the lack of domain-specific data. Ultimately, the fine-tuned LLM using this dataset demonstrates improved performance in the embodied instruction following tasks.

In addition to understanding instruction tasks with long-term decisions, the agent must consider the environment to find objects mentioned in the instruction. The mapping of instruction's targets to the objects in the environment remains a challenge, i.e., the instruction nomenclature diversity challenge. For instance, the task is '*put an apple on the table*', while common table types in the environment are: 'side table', 'dining table', and 'dresser'. The previous methods (Inoue & Ohashi, 2022; Min et al., 2022) usually predict a type of table 'side table' based on experience and plan 'find side table'. When 'side table' is not available in the environment, the agent fails to find the 'table', leading to task failure. To address this problem, we introduce environment feedback and re-planning to align environment and instruct task objects.

Not only for the monotony and simplicity of current daily tasks (Shridhar et al., 2020; Lin et al., 2023), this paper proposes RoboGPT agent for handling complex daily instruction tasks, including RoboGPT planner, Re-Plan module and RoboSkill. RoboGPT, the planning module, combines the world knowledge of LLM and the expert knowledge of robots through fine-tuning and enhancing Llama (Touvron et al., 2023a) on the collected 67K robotic dataset. It can handle hundreds of complex daily tasks and provide the proper sub-goals, some of which even require more than thirty sub-goals, e.g., '*making coffee and cleaning fruits*'. Notably, RoboGPT 1) can understand the prefix content to modify the planning sub-goals according to the environments; 2) can understand the object dependencies to find invisible objects which is in containers, e.g., '*put the apple from microwave into the garbage*', RoboGPT plans '*find the microwave first*', while current methods directly find apples, failing to find them always; 3) understand object quantities to handle tasks with more than two objects beyond other methods. Furthermore, we create a low-computational Re-Plan module to check if the environment object name match sub-goals and find equivalent replacement objects, such as 'table' replaced by 'dining table'. The Re-Plan module addresses the instruction task's nomenclature diversity challenge by adapting planning to the environment. Additionally, we improve Roboskill's navigation and interaction abilities by integrating the Fast SAM module (Zhao et al., 2023). To sum up, our contributions in this paper are mainly as follows:

- We propose a LLMs-based planner, RoboGPT, with the new robotic dataset, which combines the LLMs' common sense with the robotics expertise knowledge. As far as we known, the robotic dataset collected by us is the first high-quality, large-scale, daily task planning dataset. After fine-tuning on the new dataset, RoboGPT with strong generalization can plan hundreds of daily work tasks (even finding invisible objects in containers) and replan based on the environment, surpassing ChatGPT and other planning methods.
- A Re-Plan module is developed with low computational needs to enable the planning process to dynamically adapt to the environment, addressing the nomenclature diversity difficulty in instruction tasks. Re-Plan module receives more precise environmental information providing by the perception model integrated Fast SAM provides, improving the task execution success rate on ALFRED tasks.
- The RoboGPT agent demonstrates superior performance compared to the state-of-the-art (SOTA) models on both the ALFRED benchmark and tasks involving generalization.

## 2 RELATED WORK

Languages instruction-based policy has been a popular research area in robotics, e.g., visual language navigation, visual question and answering, and daily tasks (XU2; Shridhar et al., 2020). This paper addresses a long-term daily tasks that require navigation and interaction. Hierarchical planning methods, which use rules or expert guidance to create plans and execute low-level policies to achieve them, have proven effective (Min et al., 2022; Inoue & Ohashi, 2022; Song et al., 2023). These methods assume the agent has domain knowledge or expert assist in a closed universe. Planning issues are often described using PDDL or answer set programming in recent works (Brewka et al., 2011). A rule-based, search-based or sampling-based planning algorithm has worked for mo-

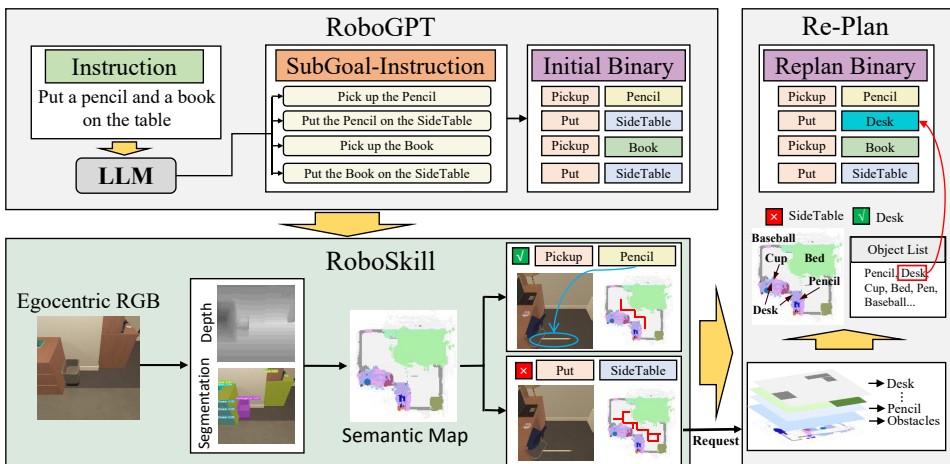

Figure 1: RoboGPT agent system. RoboGPT planner breaks a task instruction into logical sub-goals. RoboSkill will navigate, manipulate, and interact with the environment sequentially based on sub-goals. If a sub-goal fails, the Re-Plan module receives feedback and generates a new plan based on environment data, e.g., replace 'table' with 'desk'.

bile robots (Segovia-Aguas et al., 2021; Ding et al., 2020), and robotic manipulators (Cohen et al., 2010; Garrett et al., 2020). However, they use specialist-created symbolic and logical representations of planning, limited in generalizability and cannot handle unforeseen scenarios. Scholars are investigating language model-based planners (LLMs) as planning systems to address weak generality (Zeng et al., 2023; Singh et al., 2023; Ding et al., 2023). For instance, prompt engineering methods discover and extract robot-friendly planning (Yang et al., 2023; Vemprala et al., 2023). Some planning methods use a procedural language for LLMs (Liang et al., 2023), While the planning is made in an open-loop process without the world information (Huang et al., 2022; 2023; Liang et al., 2023). Saycan (Ahn et al., 2022)and Text2Motion (Ahn et al., 2022; Lin et al., 2023) utlize LLM to predict the sub-goals and select feasible actions based on environment or geometric constraints. LLMs can adjust the robot's plan online by calling LLMs multiple times (Song et al., 2023). However, the findings suggest that LLMs performance in long-term task planning is frustrating, limited with feasibility and correction, even in seemingly uncomplicated tasks. In this paper, LLM-based planning is improved with a new robotic dataset and re-planning to increase feasibility and planning correctness.

## 3 RoboGPT Agent System

A RoboGPT agent system is proposed for daily instruction tasks, including RoboGPT planner, Re-Plan module and RoboSkill (show Figure 1). Given a task instruction, RoboGPT planner decomposes it into logical sub-goals. RoboSkill performs navigation or manipulation skills based on sub-goals, produces actions that interact with the environment, and finishes all sub-goals sequentially. If a sub-goal is not completed, the Re-Plan module receives the feedback and generates a new plan based on the data received from the environment. Two key points need to be noted:

- Beyond the current monotonous and simple daily tasks, RoboGPT agent builds a more complex and diverse dataset, and the RoboGPT (a task planner with common-sense awareness of daily tasks and robot expertise) is trained to solve the long-term decision challenge, even finding invisible objects in containers.
- Due to the nomenclature diversity between the instruction and the environment, the same object can have multiple linguistic representations. The Re-Plan maps multiple instruction target representations to environment objects to solve the nomenclature diversity challenge. Moreover, a more accurate perception module is designed to work with Re-Plan to provide feasible re-planning.

### 3.1 RoboGPT Planner

Although LLMs already demonstrate remarkable performance in general-purpose scenarios. LLMs still underperform in vertical domains like robotics due to a lack of domain expertise and data. Within an academic budget, training a LLM in a specialized domain requires a strong pre-trained

LLM and high-quality training data. Therefore, this paper builds a 67K high-quality robot planning dataset and chooses the Meta's strong Llama model, to design a planning model for handling daily tasks with long-term decisions.

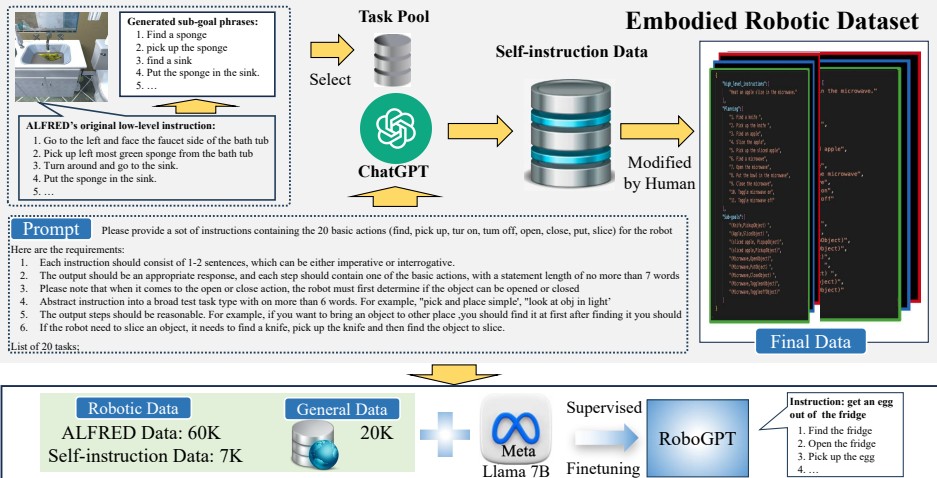

Figure 2: Framework of self-instruction data generation and RoboGPT training.

Public data for robots with long-term decisions daily tasks' planning is scarce. The most relevant ALFRED task (Shridhar et al., 2020) contains just 8K expert trajectories for 7 types of tasks, which is insufficiently diverse. As a result, we not only extract 60K samples of various types from the ALFRED data, but we also employ self-instruction to produce 7K samples with a greater variety of task descriptions and types.

**ALFRED's data:** In ALFRED task, 8k expert trajectories contain high-level instructions (task instructions), and low-level instructions (planning descriptions), e.g., a high-level instruction is: '*Put a clean sponge on a metal rack*', and the low-level instructions are: '*Go to the left and face the faucet side of the bath tub. Pick up left most green sponge from the bath tub. Turn around ... left of the lotion bottle*'. The low-level instructions carry much scenario-specific environmental information, making migration to another scenario difficult. Thus, we turn the detailed low-level instructions into sub-goal phrases. After analysis, we abstract each task's consistent planning into a single series of sub-goals. e.g., rewrite the above low-level instructions as '*find a sponge; pick up the sponge; find a sink; put the sponge in the sink; ... put the sponge on the metal rack*'. The planned sub-goals are all combinations of robot skills like navigation, grasp, open, put and so on. Thus, it can be applied to any robot and task scenario. From the 7 types of ALFRED tasks, we deduce 5 new tasks and construct planning templates to enrich robot planning data, generating a total of 60K samples.

**Self-instruction data:** The data from ALFRED demonstrations has the limitations of insufficient diversity (only 7+5 types) and low quality, which hinder the generality of the tuned model. The low quality data is due to the fact task instructions are obtained by crowdsourcing 3 individuals to watch videos, and roughly 20% of the descriptions are inaccurate (Shridhar et al., 2020). While human-written instruction data is also limited in quantity, diversity and creativity. Following the Self-Instruct(Wang et al., 2023), we generate instruction, and sub-goal samples from LLMs, then filter invalid or similar samples, modify them manually, and then use them to fine-tune the original model, seen Figure 2.

In this paper, we design precise prompts that allow ChatGPT to transform instructions into robot-performable sub-goals in a few-shot setting. In prompts, the agent should meet the common requirements of many real-world applications, e.g., output steps should be reasonable for a robot to act on.We manually select 360 samples from ALFRED's data and add them to the task pool. Here's how the data is generated: Step 1: Randomly select 20 samples from the task pool. The few-shot prompt includes these samples and the custom prompt. Step 2: ChatGPT accepts the few-shot prompt and generates divergent samples. Step 3: Regular keyword matching and duplicate checking remove redundant data. In the end, ChatGPT produces 10K samples by self-instruction. More than 50% of these samples, however, exhibit logical planning issues. We carefully review and fix the data as well, resulting in 7,274 samples that cover a variety of previously unseen daily tasks, e.g., '*I have

*a raw meat in hand, how to make a steak, put it on the table?*'.The length of task instructions and the length of sub-goals are shown in Appendix A.1: Figure 5.

**RoboGPT Planning:** We add a portion of the network's generalization data to the robot's training set and train just two episodes to help RoboGPT learn the robot's planning and keep the strong generalization ability. After training, RoboGPT can plan basic and logically difficult tasks like moving invisible objects which is in containers. For example, given the task '*Put the apple from fridge to table*', RoboGPT lets the agent find the fridge first, while current methods directly find apples, failing to find them always. Moreover, RoboGPT can also plan tasks in other domains, e.g., '*how to write a paper', 'how to learn English*', more samples shown in Appendix A.1: Table 4.

Given a task $l$, **RoboGPT** planner generates $N$ sub-goal instructions $\{P_n\}_1^N$, which are translated to robot-friendly sub-goals $S_n$ using a Mapping module **Map**():

$$\{P_n\}_1^N = \textbf{RoboGPT}(L), S_n = \textbf{Map}(Pn), S_n = (O_n/R_n, A_n), n = 1, 2, 3, ...N \qquad (1)$$

where $S_n$ contains the object $O_n$ or the receptacle $R_n$ and an action $A_n$.

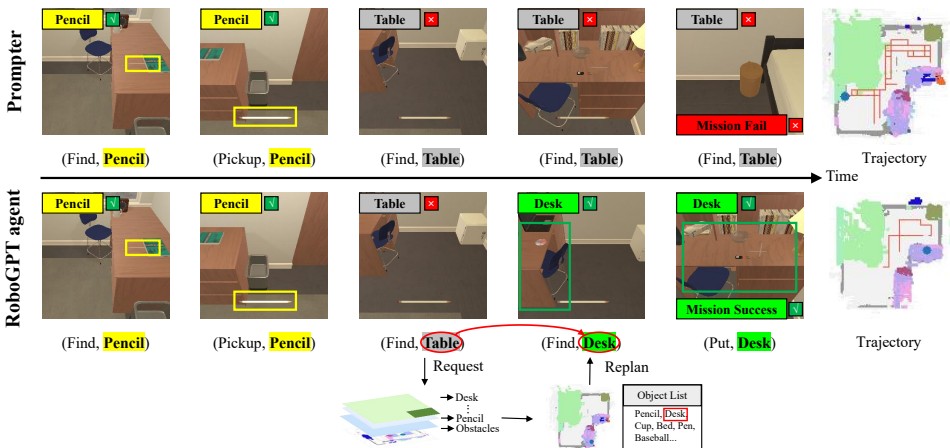

Figure 3: Effectiveness of the Re-Plan module. Re-Plan module will find similar alternative objects if the sub-goal's object cannot be noun-aligned with the object already present in the environment.

## 3.2 RE-PLAN

One of the main challenges of embodied AI is the nomenclature diversity, where the sub-goal in instructions and the item in the environment may have different names for the same object. For example, the table mentioned in the instruction is referred to as a 'coffee table' in the environment. However, existing approaches (Inoue & Ohashi, 2022) often neglect the particular issue, and they primarily focus on task planning prior to execution without fully comprehending the current environment. When there is a mismatch between the objects specified in the sub-goal and the objects present in the environment, the agent will endlessly explore until it exceeds the maximum number of steps.

LLMs-based planners (Song et al., 2023; Singh et al., 2023) usually input a list of all detected objects into the LLMs and provide a new plan, which relies on consuming LLMs and image detection, resulting in wasted resources and incorrect re-planning results once detection fails.

Different from LLMs-based planners, RoboGPT relies on semantic maps for re-planning. During the task, the semantic map is continuously updated, which can eliminate the influence of object detection errors to a certain extent. The discrete semantic voxel map $\hat{V}_t^S$ generated by depth map $I_t^D$ and semantic segmentation $I_t^S$ is combined with the binary observational map $\hat{V}_t^O$ to update the current global semantic voxel map $V_t^S$. Finally, the inventory vector $\textbf{V}_t = ObjectDetector\left(V_t^s\right)$ is obtained from $V_t^S$. Updating $V_t^S$ can eliminate the effects of object detection errors, but this correction is not timely. The global semantic map $V_t^S$ will only update the neighborhood point $V_{neighbor}$ around the target object $v_{target}$ when the agent does not detect $v_{target}$ in the target area. This means that if re-planning occurs before the semantic map $V_t^S$ is updated, RoboGPT agent could make the same mistake as other methods.

$$I_t^P = U - net\,(I_t)\,, I_t^S = SegmentationModule\,(I_t) \tag{2}$$

$$\hat{V}_t^S, \hat{V}_t^O = MapModule\left(I_t^P, I_t^S\right) \tag{3}$$

$$V_t^S = MapUpdate(\hat{V}_t^S, \hat{V}_t^O) = \hat{V}_t^S \times \hat{V}_t^O \times V_{neighbor} + V_{t-1}^S \times (1 - V_{neighbor})\,,$$
$$if\ V_{neighbor} \cap \hat{V}_t^O \neq \varnothing\ and\ v_{target}\ not\ in\ V_{neighbor} \tag{4}$$

Therefore, to address this particular scenario, RoboGPT employs the number of pixels $n_{v_i}$ as a criterion to determine whether it can be reliably confirmed that the object exists in the current scene, and gets the confirmed inventory vector $\mathbf{V}_t'$. At last, the Re-Planning module will employ a Bert model **BERT** to calculate the similarity between the target $v_i'$ and the object seen set $\mathbf{V}_t'$.

$$\mathbf{Sim}\left\{s_1, s_2, ...., s_n\right\} = \mathrm{BERT}\left(v_i', \mathbf{V}_t' = \{v_1, v_2, ...., v_n\}\right) \tag{5}$$

The most similar target $(v_j, j = argmax(\mathbf{Sim}))$ that satisfies the value $s_j > 0.7$, is considered to be the same object as $v_i'$. Replace $v_i'$ with $v_j$ to form a new sub-goal task, and the robot performs the new one. The algorithm of Re-Plan is shown in Appendix A.2: Algorithm 1.

### 3.3 ROBOSKILL

RoboSkill module completes the corresponding navigation or operation skills according to the received instructions. Referenced to Prompter (Inoue & Ohashi, 2022), RoboSkill consists of three modules: a perception module, navigation module and interaction module). At each time step, the perception module receives an egocentric RGB image, detects the position and depth maps of objects, and updates the semantic map of the corresponding positions. The semantic map guides the navigation module to look for the target. Once it finds the target, the interaction module produces actions to interact with environment until all sub-goals are completed.

Semantic segmentation plays a crucial role in various applications due to its ability to provide object masks for interaction and facilitate the creation of semantic map. The semantic map are subsequently utilized for tasks such as Re-Plan and object navigation. It has been observed that earlier models may have omissions or misidentifications, which significantly impact the correctness of the semantic map. Consequently, this leads to a decrease in the performance of Re-Plan and the overall success rate. We gather a dataset from ALFRED (Shridhar et al., 2020) seen scenes to train a semantic segmentation model utilizing the Fast SAM (Zhao et al., 2023) backbone. This significantly enhances the accuracy of both object detection and the resulting semantic map.

The shape of the semantic map is $(2 + C) \times M \times M$, where $C$ indicates the number of objects and $M \times M$ denotes the number of grids and each grid represents a 5 cm $\times$ 5 cm ground space. We make a dynamic selection of $C$ and include all large objects in the semantic map. However,for small objects, we only consider those that are relevant to the goal and the ones that can be utilized for Re-Plan purposes. For instance, if the goal is to 'Place a bottle on the desk' and the sub-goal is to 'Find a glass bottle', we would include objects such as 'glass bottle', 'wine bottle', 'soap bottle', and other types of bottles in the semantic map.

## 4 EXPERIMENTS

**Metrics:** Success rate (SR), goal-condition success (GC), and high-level planning accuracy (HLP ACC) are reported. SR is the agent's over all tasks completion rate. GC is the ratio of completed goal-conditions, e.g., in 'Heating a cleaned apple', 'washing' and 'heating' are goal-conditions. Using SR and GC, the path length weighted SR (PLWSR) and path length weighted GC (PLWGC) are defined as (path length of the expert trajectory)/ (path length taken by the agent). HLP ACC is the accuracy of sub-goals planning.

**Baselines:** We choose the most effective methods for the ALFRED daily tasks: end-to-end models with step-by-step instructions, and high-level instruction-only approaches. Prompter, which plans

using templates, and LLM-Planer, which plans using ChatGPT, are the methods that are most similar to ours; for details on their particular experimental settings, see the Appendix A.4 A.3.

**Evaluation Dataset:** The test set has four parts: 'Tests Seen' (1533 episodes), 'Tests Unseen' (1529 episodes) and 'Valid Unseen' (50 episodes) in ALFRED, as well as our own generated 'Generalization Task' (50 episodes). Although ALFRED has ground truth for each valid tasks, we find some of them have issues (see Appendix B.1: Figure 8). Therefore, we select 50 tasks from 'Valid Unseen' where the instructions and ground truth are perfectly matched. The 'Generalization Task' consists of 50 complex high-level instructions that we annotate ourselves, and these tasks fall completely outside the seven categories defined by **ALFRED**.

**Training Details:** RoboGPT planner is fine-tuned from Llama-7b on NVIDIA DGX A100. Training data includes 67K proprietary and 20K online generic data. To maintain generalization, network is trained in 2 episodes with a $10^{-5}$ learning rate. RoboSkill trains a semantic segmentation model leveraging the Fast SAM backbone (Zhao et al., 2023) on collected 80k images for 100 epochs with a learning rate of $10^{-3}$, batch size is 16.

## 4.1 EXPERIMENTAL RESULTS

| Method | Valid Unseen | | | | | Gen. Task |
|---|---|---|---|---|---|---|
| | SR | PLWSR | GC | PLWGC | HLP ACC | HLP ACC |
| High-level Instruction Only | | | | | | |
| Prompter (Inoue & Ohashi, 2022) | 50 | 20.41 | 56.67 | 21.86 | 82 | 0 |
| LLM-Planner (Song et al., 2023) | 32 | 13.54 | 47.33 | 18.74 | 66 | 34 |
| **RoboGPT agent** | **60** | 19.10 | **69.83** | 21.26 | **96** | **78** |
| **RoboGPT agent w.o. RoboGPT** | 58 | **22.18** | 65.83 | **24.17** | 58 | 0 |
| **RoboGPT agent w.o. RoboSkill** | 56 | 21.80 | 64.00 | 23.51 | **96** | **78** |
| **RoboGPT agent w.o. Re-Plan** | 52 | 16.02 | 62.33 | 19.01 | 88 | 78 |

Table 1: Performance comparison on 7 types of ALFRED tasks (Valid Unseen) and Generalization tasks (Gen. Task)

Table 1 summarizes the performance of our RoboGPT and the SOTA methods: Prompter (Inoue & Ohashi, 2022) with template-based planner and LLM-Planner (Song et al., 2023) with ChatGPT planner. The RoboGPT achieves 10.00% absolute (20.00% relative) gain in SR (success rate) than the SOTA method Prompter on Valid Unseen data. Our RoboGPT also achieves a definite advantage in HLP ACC (the accuracy of task planning): 14.0% absolute (17.0% relative) gain on 7 types of tasks, significantly exceeding other methods.

Notably, facing the Generalization tasks beyond the 7 type tasks, RoboGPT outperforms LLM-Planner (Song et al., 2023) by 40%, and outperforms the Prompter (Inoue & Ohashi, 2022) by 78%! This implies that RoboGPT is extremely generalizable and capable of daily instruction tasks of long-term decisions.

RoboGPT agent without RoboGPT planner has the highest PLWSR and PLWGC than others, even for the one using RoboGPT. The reason is Prompter combined some objects with similar meanings (such as desk lamp and floor lamp, butter knife and knife) based on rules. Although this can significantly improve the execution efficiency of the agent, it is difficult to transfer to complex real scenes. Contrary to Prompter, RoboGPT uses the Re-Plan module to fix the target objects but only after gathering a significant amount of environmental data. Although this strategy is more useful in the actual world, it can also decrease the robot's execution efficiency, resulting in lower scores for PLWSR and PLWGC.

## 4.2 ABLATION EXPERIMENTS

**Effectiveness of RoboGPT planning:** In the third-to-last column of Table 1, RoboGPT agent's planning module is replaced with the Prompter's planner (Inoue & Ohashi, 2022), resulting in a 8% decrease in SR and a large decrease in HLP ACC, suggesting that RoboGPT has a crucial role to play in understanding the tasks with long-term decisions than Prompter. In designed Generalization task RoboGPT performs well in complicated long-term tasks, as shown by its HLP ACC in Table 1 being far ahead of others. Prompter performs effectively on ALFRED tasks because it has a template intended for ALFRED and does not generalize to other tasks. LLM-Planner employs ChatGPT

for logical reasoning, however its performance depends on pre-labeled data for prompt. While exhibiting a certain level of generalization, the LLM-Planner's performance remains unstable.

**Analysis of planning results:** The Prompter and FILM (Inoue & Ohashi, 2022; Min et al., 2022) view task's planning as a classification task and then use a template to plan. They may misjudge the task type, the parent target and the target object when planning a task, e.g., the task is '*put two potatoes in microwave*', while Prompter plans '*pick up an egg*'. It could be due to overfitting the training data (some examples shown in Appendix B.1: Figure 6). The LLM-Planner (Song et al., 2023) uses ChatGPT to plan and may make logical errors when planing; e.g., for the task '*Place a glass with a knife in a sink.*', it will plan '*pick up glass then pick up the knife, and put knife in the sink*', which doesn't put the knife in the glass, failing to understand the relationship between object and container (some examples shown in Appendix B.1: Figure 7).

Beyond the above methods, the advantages of RoboGPT can be summarized as follows: 1) **prefix understood**: using environment information as a prefix prompt, it can produce practical planning; e.g., for the task '*There is a stove and no microwave, how to heat an apple*', RoboGPT plans to '*find a stove*' to heat the apple, while others usually plan to '*find a microwave*'. 2) **quantity understood**, RoboGPT understands that robots must pick up numerous objects one by one, not all at once, and can plan tasks with 3 or more objects, while other approaches can only plan tasks with 2 objects. e.g., '*put four books on the desk*' 3) **object dependencies understood**, RoboGOPT can use task instructions to infer the location of invisible objects. e.g., if the task is 'put the apple in the fridge and then on the table', RoboGPT will plan to 'find the fridge', first, whereas others will simply find an apple (shown in Figure 4). 4) **tasks with ultra-long-term decision understood** RoboGOPT can understand and plan complex tasks with more than 30 sub-goals. e.g., '*cut a slice of bread, warm it with the microwave, put it on the counter along with putting the knife in the cabinet*'. More examples are shown in Appendix A.1 Table 4.

**Instruction:** Get a towel out of the cabinet to soak it, then put it on the toilet

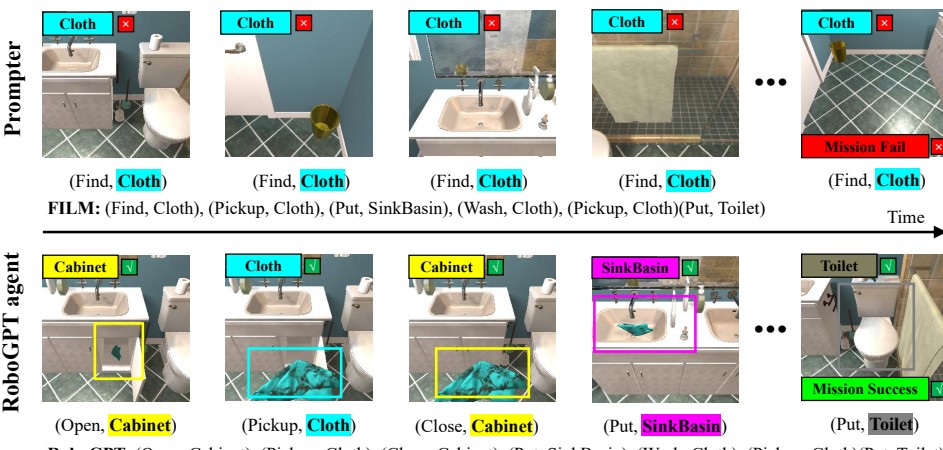

Figure 4: Planning of the task with invisible object in container. RoboGPT possesses the capability to comprehend object relationships, hence facilitating the accomplishment of tasks involving the presence of a target object within an enclosed area.

**Effectiveness of Re-Plan:** We employ Re-Plan module with low-computing to achieve quick replanning. Compared with RoboGPT agent without Re-Plan in the last row of Table 1, RoboGPT agent achieves higher PLWSR and SR, and is able to finish the task in the AI2THOR environment with fewer steps resulting in higher PLWSR. When the agent cannot find the target, Re-Plan would replace the target with the similar object in semantic map, e.g., replace 'dinning table' with 'side table'. It proves that Re-Plan helps with understanding the environment and the alignments of environment objects and targets in instructions.

**Effectiveness of RoboSkill:** When comparing RoboGPT agent without RoboSkill in the last second row of Table 1, RoboGPT agent produces significantly higher SR (+4% improvement) and GC (+5.83% improvement), which indicates a major improvement in perception. As a result, RoboSkill can build an accurate semantic map and couple it with Re-Plan, leading to further development. The segmentation and detection of RoboSkill greatly outperform those of others (see the Appendix B.4 for details). Additionally, although the success rate of RoboGPT planning (HLP ACC) and the

(goal-condition succes) GC are high, the success rate of execution is not that high, indicating that many tasks fail during the interaction. As a result, the interaction algorithm can be enhanced going forward to increase the robot's success rate in carrying out tasks.

## 4.3 EXPERIMENTAL RESULTS ON ALFRED TEST SETS

| Method | Low Inst. | LLM | Tem. | Tests Seen | | Tests Unseen | |
|---|---|---|---|---|---|---|---|
| | | | | GC | SR | GC | SR |
| HLSM (Blukis et al., 2021) | ✓ | ✗ | ✗ | 41.21 | 29.94 | 30.31 | 20.27 |
| LGS-RPA (Murray & Cakmak, 2022) | ✓ | ✗ | ✗ | **48.66** | **40.05** | **45.24** | **35.41** |
| ET (Pashevich et al., 2021) | ✓ | ✗ | ✗ | 45.44 | 38.42 | 18.56 | 8.57 |
| MCR-Agent Bhambri et al. (2023) | ✓ | ✗ | ✗ | - | 30.13 | - | 17.04 |
| M-TRACK Song et al. (2022) | ✓ | ✗ | ✗ | 22.60 | 16.29 | 33.35 | 24.79 |
| LEBP Liu et al. (2022a) | ✓ | ✗ | ✗ | 36.79 | 28.30 | 36.33 | 28.97 |
| EPA (Liu et al., 2022b) | ✗ | ✗ | ✗ | 44.14 | 39.96 | 39.54 | 36.07 |
| HLSM (Blukis et al., 2021) | ✗ | ✗ | ✗ | 35.79 | 25.11 | 27.24 | 16.29 |
| FILM (Min et al., 2022) | ✗ | ✗ | ✓ | 36.15 | 25.77 | 34.75 | 24.46 |
| Prompter (Inoue & Ohashi, 2022) | ✗ | ✗ | ✓ | 55.90 | 49.38 | **59.55** | 42.64 |
| **RoboGPT agnet (Ours)** | ✗ | ✗ | ✓ | **58.39** | **49.45** | 55.01 | **44.57** |
| LLM-Planner (Song et al., 2023) | ✗ | ✓ | ✗ | 26.77 | 18.20 | 23.37 | 16.42 |
| **RoboGPT agnet (Ours)** | ✗ | ✓ | ✗ | **54.99** | **45.66** | **53.80** | **42.97** |

Table 2: Results of ALFRED tasks. **Bold** numbers are top scores in each section, and experimental results of all methods come from the official **ALFRED** ranking or paper. 'Low Inst.', 'LLM' and 'Tem.' respectively represent the utilization of low-level instructions, LLMs, and templates. The proposed RoboGPT agent primarily focuses on the decomposition of high-level instructions into sub-goals, aligning with the practical working environment of household robots.

We perform our method on ALFRED test sets and achieve SOTA results on seen and unseen (shown in Table 2). Compared to methods requiring low-level step-by-step instructions (Bhambri et al., 2023; Murray & Cakmak, 2022) and the method requiring ChatGPT planning (Song et al., 2023), RoboGPT agent achieves the best performance with at least 9.16% improvement of SR in Test Unseen, showing the plan ability of RoboGPT agent even beyond the step-by-step guidance of an expert or ChatGPT. Additionally, RoboGPT agent only marginally improves when compared to the template-based method (Inoue & Ohashi, 2022). The main reason for this is that ALFRED is a crowdsourced form of labeled data, with many irregularities in the labeling. By our count more than 20% of the high-level instructions (instruction tasks) are ambiguous or even incorrect (the detail analysis and examples shown in Appendix B.3 and Figure 9). Wrong instruction tasks lead to errors in our planning. This explains why there is a difference in SR of different methods between Table 1 and Table 2. Prompter categorizes tasks first and plans depending on the categories using the template. Due to overfitting, it may correctly categorize some error instructions for correct types, that produce proper planning results. Although some complex tasks are accurately planned using our method, carrying out complex tasks is challenging, which leads to a slightly greater success rate.

**Verification System:** We develop a verification system based on the Ai2Thor simulation system. You can input arbitrary natural language commands and use the RoboGPT agent to interact with the environment to accomplish the task. The demo shows in Appendix B.5: Figure 10 and supplementary Material.

## 5 CONCLUSIONS

In this paper, we design a RoboGPT agent for solving daily instruction tasks with long-term decisions. The planning module RoboGPT enhances and fine-tunes Llama using the collected 67K robotic dataset to integrate the world knowledge of LLMs with the expert knowledge of robots, which can understand the prefix context, object quantities, object dependencies and the tasks with long-term decisions, handling most of daily tasks. The designed Re-Plan module adapts the planning to the environment, mitigating the nomenclature diversity problem. Additionally, a Roboskill with an accurate perception model Fast SAM is developed, resulting in improved navigation and manipulation abilities. The paper provides a well-generalized method for daily instruction tasks with long-term decisions in robotics. Future improvements may focus on multi-modal robot planning and manipulation.

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

# A APPENDIX

## A.1 DATASET

The collected data involves 60K ALFRED's data and 7K self-instruction data. ALFRED's data is generated from the ALFRED tasks, (which includes 8K expert trajectories for 7 types of tasks), and self-instruction data is generated by ChatGPT and modified by humans. In Figure 5, we also analyze the length of task instructions and the length of sub-goals. The figure shows that the data we constructed includes many complex tasks with more than 15 sub-goals and many tasks whose instructions are more than 20 letters long.

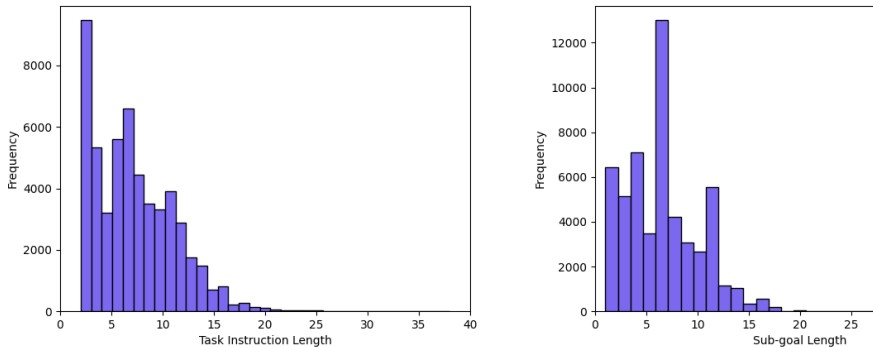

Figure 5: Distribution of the length of task instructions. Distribution of sub-goal lengths for generated data (right). The data contains some complex tasks with morn than 15 sub-goals.

## A.2 PSEUDOCODE FOR THE RE-PLAN

Using the objects in the semantic map as environmental information, the Re-Plan module will use **BERT** to compute the similarity between the target $v_i^{'}$ and the set of objects seen $\mathbf{V}_{\mathbf{t}}^{'}$. The similarity is used to determine the object to be replaced. The pseudocode of Re-Plan is shown in Algorithm 1.

---
**Algorithm 1:** RoboGPT Re-Plan module
---
1   $t \leftarrow 0$; $I \leftarrow$ Instruction; $N \leftarrow$ Pixel threshold;
2   SubTask list $S \leftarrow$ RoboGPT($I$);
3   **while** *not done* **do**
4      $I_t \leftarrow$ *getObservation*();
5      $I_t^D \leftarrow$ *U-net*($I_t$);
6      $I_t^S \leftarrow$ *SegmentationModule*($I_t$);
7      $V_t^S \leftarrow$ *MapModule*($I_t^D, I_t^S$);
8      **if** *no target found after $n$ steps* **then**
9          $V_t^S \leftarrow$ *MapUpdate*($\hat{V}_t^S, \hat{V}_t^O$);
10         $\mathbf{V}_t \leftarrow$ *Object-Detector*($V_t^S$);
11         $\mathbf{V}_t' \leftarrow [v_i$ for $v_i$ in $\mathbf{V}_t$ if $n_{v_i} > N$ ];
12         $\mathbf{Sim} \leftarrow$ BERT($\mathbf{V}_t', S$);
13         $S' \leftarrow$ *SubgoalUpdate*($\mathbf{Sim}, S$);
14         $t \leftarrow t + 1$;
15      **else**
16         $t \leftarrow t + 1$;
17      **end**
18 **end**
---

## A.3   PROMPTER

Prompter (Inoue & Ohashi, 2022) has been enhanced based on the FILM (Min et al., 2022) framework, which consists of three components: high-level planner, perceptual mapping, and navigation/interaction strategy. The high-level planner used by Prompter is similar to the one employed by FILM. It employs a classifier based on BERT to categorize each task into one of seven task types. Additionally, it predicts parameters associated with each task, including target objects, receptacles, parent objects, and slicing. One notable distinction between Prompter and FILM lies in their respective semantic search modules. Prompter search goal objects based on landmarks, which speeds up the search. In the comparative experiment, the pre-generated sub-goals provided by Prompter are utilized without any modifications. In contrast to LLM-Planner Song et al. (2023), which utilizes a limited dataset to fine-tune BERT in order to generate FILM's few-shot results.

## A.4   LLM-PLANNER

The LLM-Planner (Song et al., 2023) consists of two main components, namely the high-level planner and the replan module. These components are integrated into the overall framework. Thus, in the comparative experiment, the high-level planner based on Prompter's template was substituted with LLM-Planner. A total of 140 cases, 20 for each of the seven ALFRED tasks, were chosen. To assess the similarity between each training example and the current test example, the pre-trained BERT-base-uncased model was employed. The concept of similarity was established by calculating the Euclidean distance between BERT embedding. The k-nearest-neighbor (KNN) retriever is employed to select the nine most comparable in-context cases. These examples are then fed into the LLM to generate a static version of LLM-Planner. In order to reduce the frequency of LLM calls, we provide LLM-Planner with an inequitable advantage by supplying it with all scene objects prior. The prompt of LLM-Planner is shown in Table 3.

Table 3: Examples of LLM-Planner prompts

---
Create a high-level plan for completing a household task using the allowed actions and visible objects.

Allowed actions: OpenObject, CloseObject, PickupObject, PutObject, ToggleObjectOn, ToggleObjectOff, SliceObject, FindObject

Task description: put the pen on the desk
---

Table 3: Examples of LLM-Planner prompts continued

| |
|---|
| Visible objects are Bed, coffee, Book, DeskLamp, Mug, Shelf, Pen, Desk Next Plan: FindObject Pen, PickupObject Pen, FindObject Desk, PutObject Desk |
| Task description: put the spray bottle on the toilet |
| Visible objects are SinkBasin, Cabinet, Pillow, SprayBottle, Toilet, Cloth Next Plan: FindObject Spraybottle, PickupObject Spraybottle, FindObject Toilet, PutObject Toilet |
| . . . |
| Task description: Pick up a coffee mug from the desk and put it down on the desk |
| Visible objects are Mug, Desk, DeskLamp, Pencil, Book |
| Next Plan: |

# B  ADDITIONAL RESULTS ANALYSIS

## B.1  ANALYSIS OF PLANNING RESULTS

Prompter (Inoue & Ohashi, 2022) may misjudge the task type, parent goal, and target object when planning a task, as shown in Figure 6), potentially disregarding valuable information. For instance, in the instruction 'put the apple from the fridge on the table', Prompter may overlook the crucial detail that the apple is located inside the fridge. Furthermore, Prompter utilizes a predefined template, which is limited to specific types of tasks. LLM-Planner (Song et al., 2023) employs ChatGPT as its planner, which can be confused in logic, miss some actions, ignore robot's constraints, etc. (refer to Figure 7). In contrast to RoboGPT, LLM-Planner is less intelligent, when instructed to clean three apples and put them on the table, LLM-Planner cleans each apple separately, but RoboGPT cleans them together. RoboGPT demonstrates a high level of cognitive ability, as it is capable of comprehending prefix content, discerning object amounts, understanding object dependencies, and making long-term decisions, as shown in Table 4.

> **Examples of Mistake in FILM/Prompter**
>
> **Instruction:** Put a cup on the desk, by the chair.
> **Output:** **Pick up Pencil**, Put the pencil in the Cup, Pick up Cup, Put the Cup on the Desk
> **Error analysis:** Misestimate the task type (pick_and_place_simple → pick_and_place_with_movable_recep).
>
> **Instruction:** Put two bottles of soap on the **shelf** left of the tissue box.
> **Output:** Pick up Soapbottle, Put the Soapbottle on the **Cart**, Pick up Soapbottle, Put the Soapbottle on the **Cart**
> **Error analysis:** Misestimate the parent target.
>
> **Instruction:** Put two **potatoes** in a microwave.
> **Output:** Pick up **Egg**, Open Microwave, Put the egg in Microwave, Close Microwave, Pick up **Egg**, Open Microwave, Put the egg in the Microwave, Close Microwave
> **Error analysis:** Misestimate the target object.

Figure 6: Examples of Mistake in FILM/Prompter

> **Examples of Mistake in LLM-Planner**
>
> **Instruction:** Put a egg on a bowl on a counter.
> **Output:** Find Egg, **Pick up Egg**, Find Bowl, **Pick up Bowl**, Find Countertop, Putthe bowl on the Countertop
> **Error analysis:** Failure to understand the agent's executable actions.
>
> **Instruction:** Place a glass with a knife in a sink.
> **Output:** Find Glass, **Pick up Glass**, Find Knife, **Put SinkBasin**, Pick up Knife, Find SinkBasin, Put SinkBasin
> **Error analysis:** Failure to understand the relationship between object and container.
>
> **Instruction:** Put a **cooked** apple in the sink.
> **Output:** Find Apple, Pick up Apple, Find SinkBasin, Put the apple in the SinkBasin
> **Error analysis:** Failure to respond to the keyword "cooked".

Figure 7: Examples of Mistake in LLM-Planner

Table 4: Examples of RoboGPT, LLM-Planner and Prompter planning

| |
|---|
| **Prefix understanding** |
| **High-level instruction: 'There is an oven and no microwave, how to heat an apple '** |
| **RoboGPT planner**:[ Find an oven, Open the oven, Find an apple, Pick up the apple, Put the apple on oven, Close the oven, Toggle on the oven, Toggle off the oven ] |
| **LLM-Planner**:[Find a oven, Pick up an apple, Find an oven, Put the apple on oven] error: didn't open the oven before putting the apple, didn't toggle on the oven |
| **Prompter**:[ Find a microwave, Open the microwave, Find an apple, Pick up the apple, Put the apple on microwave, Close the microwave, Toggle on the microwave, Toggle off the microwave ] error: failure to understand prefixes |
| |
| **High-level instruction: 'There is no cup but a mug, how to put a cup on the desk by the chair '** |
| **RoboGPT planner**:[ Find a mug, Pick up the mug, Find a desk, Put the mug on desk ] |
| **LLM planner**:[ Find a mug, Pick up the mug, Find a desk, Put the mug on the desk, Find a chair, Put the chair near the desk ] unnecessary: no need to move chair |
| **Prompter**:[ Find a cup, Pick up the cup, Find a desk, Put the cup on desk ] error: failure to understand prefixes |
| |
| **Quantitative understanding of objects** |
| **High-level instruction: 'Heat up three apples and place them on the table '** |
| **RoboGPT planner**: [ Find an apple, Pick up the apple, Find a microwave, Open the microwave, Put the apple on microwave, Find another apple, Pick up the apple, Find the microwave, Put the apple on microwave, Find a third apple, Pick up the apple, Find the microwave, Put the apple on microwave, Close the microwave, Toggle microwave on, Toggle microwave off, Open the microwave, Pick up an apple, Find a table, Put the apple on the table, Find the microwave, Pick up an apple, Find a table, Put the apple on table, Find the microwave, Pick up a third apple, Find a table, Put the apple on table ] |

Table 4: Examples of RoboGPT, LLM-Planner and Prompter planning (continued)

**LLM-Planner**:[ Find an apple, Pick up the apple, Find a microwave, Open the microwave, Put the apple in the microwave, Close the microwave, Turn on the microwave, Turn off the microwave, Open the microwave, Pick up the apple, Close the microwave, Find a table, Put the apple on the table, Find another apple, Pick up the apple, Find the microwave, Open the microwave, Put the apple in the microwave, Close the microwave, Turn on the microwave, Turn off the microwave, Open the microwave, Pick up the apple, Close the microwave, Find the table, Put the apple on the table, Find another apple, Pick up the apple, Find the microwave, Open the microwave, Put the apple in the microwave, Close the microwave, Turn on the microwave, Turn off the microwave, Open the microwave, Pick up the apple, Close the microwave, Find the table, Put the apple on the table ] unintelligent: heat each apple individually

**Prompter**:[ Find an apple, Pick up the apple, Find the table, Put the apple on the table, Find another apple, Pick up the apple, Find the table, Put the apple on the table ] error: only put two apple and didn't heat

**High-level instruction: 'Clean three apples and put them on the table '**

**RoboGPT planner**:[ Find an apple, Pick up the apple, Find a sink basin, Put the apple on sink basin, Find an another apple, Pick up the apple, Find the sink basin, Put the apple on sink basin, Find a third apple, Pick up the apple, Find the sink basin, Put the apple on sink basin, Toggle faucet on, Toggle faucet off, Pick up an apple, Find a dining table, Put the apple on dining table, Find the sink basin, Pick up an another apple, Find a dining table, Put the apple on dining table, Find the sink basin, Pick up a third apple, Find a dining table, Put the apple on dining table ]

**LLM-Planner**:[ Find an apple, Pick up the apple, Find a sink basin, Put the apple in the sink basin, Turn on the faucet, Turn off the faucet, Pick up the apple, Find a table, Put the apple on the table, Find another apple, Pick up the apple, Find a sink basin, Put the apple in the sink basin, Turn on the faucet, Turn off the faucet, Pick up the apple, Find a table, Put the apple on the table, Find another apple, Pick up the apple, Find a sink basin, Put the apple in the sink basin, Turn on the faucet, Turn off the faucet, Pick up the apple, Find a table, Put the apple on the table ] unintelligent: wash each apple separately

**Prompter**:[ Find an apple, Pick up the apple, Find a table, Put the apple on the table, Find another apple, Pick up the apple, Find a table, Put the apple on the table ] error: didn't clean and only put two apples

**Object dependencies understanding**

**High-level instruction: 'Pick up the apple in the fridge and then put it on the table '**

**RoboGPT planner**:[ Find a fridge, Open the fridge, Find an apple, Pick up the apple, Find a table, Put the apple on table, Find a fridge, Close the fridge ]

**LLM-Planner**:[ Find the fridge, Open the fridge, Find the apple, Pick up the apple, Find the table, Put the apple on the table ] error: didn't close the fridge

**Prompter**:[ Find the apple, Pick up the apple, Find the table, Put the apple on the table ] error: can't found apple without opening the fridge

**High-level instruction: 'Microwave a mug from the cupboard and put it next to the toaster '**

**RoboGPT planner**: [ Find a cupboard, Open the cupboard, Find a mug, Pick up the mug, Close the cupboard, Find a microwave, Open the microwave, Put the mug on microwave, Close the microwave, Toggle microwave on, Toggle microwave off, Open the microwave, Pick up the mug, Close the microwave, Find a counter top, Put the mug on counter top ]

**LLM-Planner**:[ Find an microwave, Find an mug, Find an cupboard, Open the cupboard, Pick up an mug, Close the cupboard, Find an microwave, Open the microwave, Put the mug in the microwave, Close the microwave, Toggle on the microwave, Toggle off the microwave, Open the microwave, Pick up the mug, Close the microwave, Find a toaster, Put the mug on the toaster ] error: put the mug on the toaster

Table 4: Examples of RoboGPT, LLM-Planner and Prompter planning (continued)

**Prompter**: [ Find a mug, Pick up the mug, Find a microwave, Open the microwave, Put the mug on microwave, Close the microwave, Toggle microwave on, Toggle microwave off, Open the microwave, Pick up the mug, Close the microwave, Find a counter top, Put the mug on counter top ] error: can't find mug without opening the microwave

---

**Tasks with ultra-long-term decision understanding**

**High-level instruction: 'Cut a slice of bread, warm it with the microwave, put it on the counter along with putting the knife in the cabinet'**

**RoboGPT planner**:[ Find a knife, Pick up the knife, Find a bread, Slice the bread, Find a sink basin, Put the knife on the sink basin, Toggle faucet on, Toggle faucet off, Pick up the knife, Find a cabinet, Open the cabinet, Put the knife in the cabinet, Close the cabinet, Find a plate, Pick up the plate, Find the sliced bread, Put the plate on the side of the sliced bread, Pick up the sliced bread, Put the sliced bread on the plate, Find a microwave, Open the microwave, Put the plate with sliced bread in the microwave, Close the microwave, Toggle microwave on, Toggle microwave off, Open the microwave, Pick up the plate with sliced bread in, Close the microwave, Find a counter, Put the plate with sliced bread on the counter ]

**LLM-Planner**:[ Find a knife, Pick up the knife, Find some bread, Slice the bread, Find a microwave, Open the microwave, Put the bread slice in the microwave, Close the microwave, Turn on the microwave, Turn off the microwave, Open the microwave, Pick up the sliced bread, Close the microwave, Find a counter, Put the sliced bread on the counter, Find a cabinet, Open the cabinet, Put the knife in the cabinet, Close the cabinet ] error: the robot can only pick up one thing at a time, and by the time it picks up the apple, the knife has already dropped, so it should do like RoboGPT and put the to in the cupboard first

**Prompter**:[ Find a knife, Pick up the knife, Find a bread, Slice the bread, Find a sink basin, Put the knife on the sink basin, Find the sliced bread, Pick up the sliced bread, Find a microwave, Open the microwave, Put the sliced bread in the microwave, Close the microwave, Toggle microwave on, Toggle microwave off, Open the microwave, Pick up the sliced bread, Close the microwave, Find a counter, Put sliced bread on the counter ] error: didn't put the knife in the cupboard

**High-level instruction: 'Put a bowl with a cleaned lettuce in it on the table '**

**RoboGPT planner**:[ Find a lettuce, Pick up the lettuce, Find a sink basin, Put the lettuce on sink basin, Toggle faucet on, Toggle faucet off, Pick up the lettuce, Find a bowl, Put the lettuce in bowl, Pick up the bowl, Find a table, Put the bowl on table ]

**LLM-Planner**:[ Find a sink basin, Find a bowl, Put the bowl, Find the lettuce, Put the lettuce, Pickup the bowl, Find a table, Put the table ] error: logical disorder and didn't clean the lettuce

**Prompter**:[ Find a lettuce, Pick up the lettuce, Find a sink basin, Put the lettuce on sink basin, Toggle faucet on, Toggle faucet off, Pick up the lettuce, Put the lettuce on table ] error: didn't put the lettuce in a bowl

## B.2 ANALYSIS OF RE-PLAN

Re-Plan module aims to solve the mismatch of object in high-level instructions and the detected objects in environment. It will find similar alternative objects if the sub- goal's object cannot be noun-aligned with the object already present in the environment. From Table 1 we can see the model with Re-Plan is 8% higher in SR than the one without it, indicating that Re-Plan has a significant role.

The RoboGPT agent without RoboGPT planner has the highest PLWSR and PLWGC than others, even for the one using RoboGPT. The reason for this is the instruction nomenclature diversity, for instance, the task is 'put an apple on the table', while all table types in the environment are: 'side table', 'dining table', and 'dresser'. RoboGPT planner plans 'Find a table' according to the instruction,

and after finding several times Re-Plan produces a new plan 'Find a side table' based on environment feedback. Therefore, RoboGPT agent needs more steps if the instruction has some ambiguous objects. While Prompter's template planning has overfitted the data and will correspond the table directly to the most frequent 'side table' in the environment, causing it to find objects without the need for more steps, and resulting in the template not being easily migrated to other environments.

## B.3    ANALYSIS OF INSTRUCTIONS

ALFRED is a crowdsourced form of labelling data. Videos of expert trajectories are generated and humans annotate high-level instructions based on the videos. There are many irregularities and even errors in the labeling. By our count about 30% of the high-level instructions (instruction tasks) are ambiguous or even incorrect, e.g., the expert trajectory is 'pick up the mobile phone and put it under the lamp to look at it', while the annotation is 'turn on the lamp', which lacks the most crucial part of holding something under the lamp to look at it. More examples are shown in Figure 8.

For the wrongly instruction RoboGPT plans 'find the lamp, turn on the lamp', which is not right for the ground truth, resulting in the failure of the agent (shown in Figure 9).

Prompter first classifies the tasks and then plans them using templates based on the classification. Due to overfitting. it may correctly classify the wrong instruction as the correct type ('$look\_at\_object\_under\_lamp$') and thus produce correct planning results 'Find a cellphone, pick up the cellphone, find a lamp, turn on the lamp'. While it is possible to accurately plan some complex tasks using RoboGPT, performing complex tasks is challenging performing, and thus the success rate is slightly higher. This explains why RoboGPT boosts over prompter in Table 2 are not as large as Table 1.

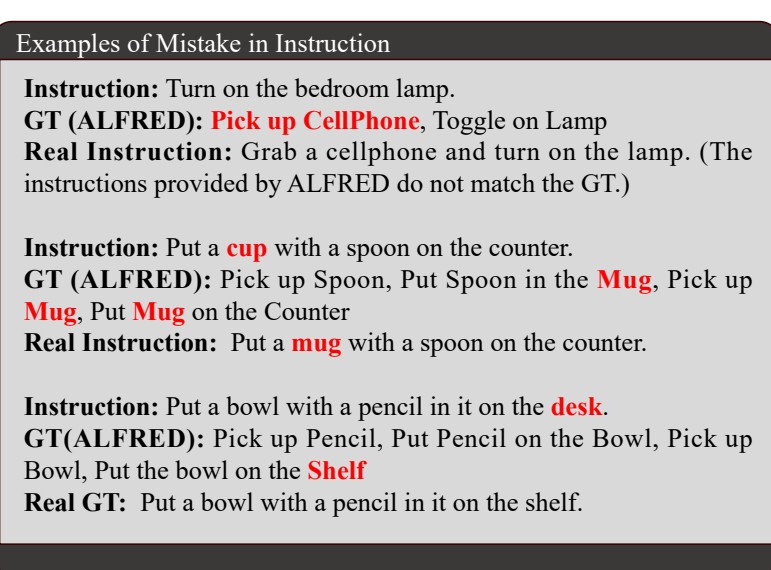

Figure 8: Examples of mistake instructions

## B.4    PERCEPTION MODEL

In order to train a semantic segmentation model leveraging the Fast SAM backbone (Zhao et al., 2023), we build a dataset comprising over 80,000 image pairs from seen scenes collected in the ALFRED dataset. Our model is built upon the pre-trained YOLOv8x-seg model and was trained on an NVIDIA DGX A100 for 100 epochs with a learning rate of $10^{-3}$, batch size is 16.

For evaluating the performance of our novel model, we re-collect more than 8,000 images from previously unseen scenes in ALFRED dataset. Subsequently, we execute each algorithm and measured their performance using two key metrics: mean Average Recall (mAR) in Intersection over Union (IoU) ranging from 0.50 to 0.95 and mean Average Precision (mAP) in IoU from 0.50 to 0.95.

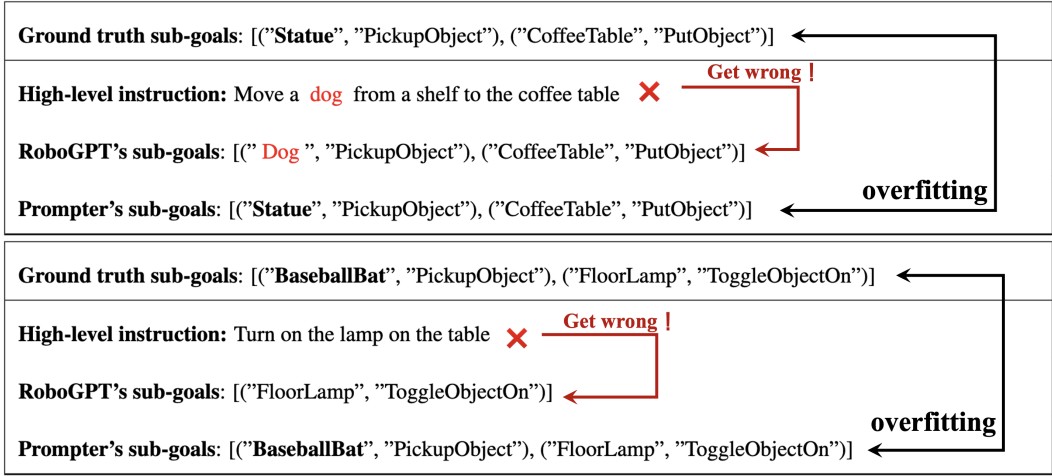

Figure 9: Examples of Prompter and RoboGPT planning with mistake instructions

Table 5: Comparison of our perception model with Mask RCNN

| Method | Bounding Box | | Segmentation | |
|---|---|---|---|---|
| | mAR | mAP | mAR | mAP |
| Mask RCNN | 0.501 | 0.658 | 0.434 | 0.667 |
| **Perception model (ours)** | 0.889 | 0.871 | 0.872 | 0.697 |

## B.5   DEMO

We develop a verification system based on the Ai2thor simulation system. You can input arbitrary natural language commands and use the RoboGPT agent to interact with the environment to accomplish the task. The demo shows in Figure 10. The instruction task is 'Slice a tomato, put the knife in the sink and put the sliced tomato in the fridge', and RoboGPT plans 'Find a knife, pick up the knife, find a tomato, slice the tomato, find a sink, put the knife in the sink, find the sliced tomato, pick up the sliced tomato, find a fridge, open the fridge, put the sliced tomato in the fridge, close the fridge'. The figure shows the key frame and the full demo can be found in the supplementary material.

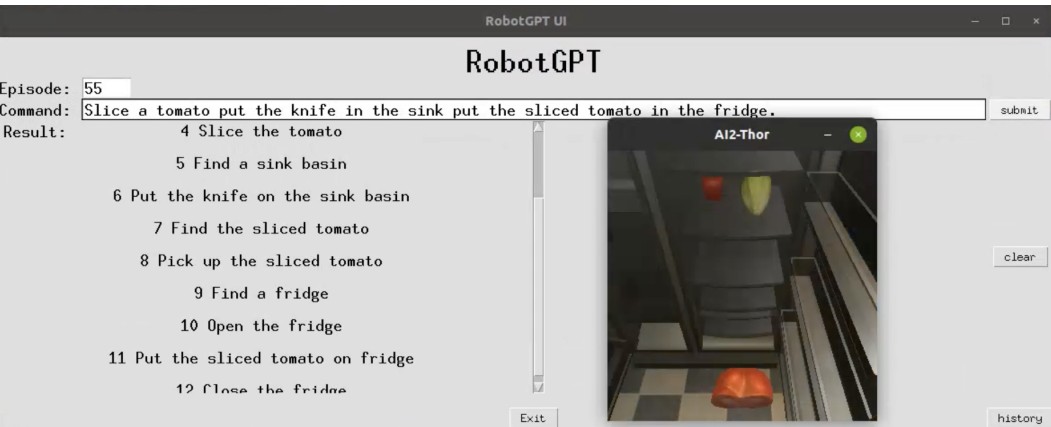

Figure 10: Verification system. The instruction task is 'Slice a tomato, put the knife in the sink and put the sliced tomato in the fridge'. The planning of RoboGPT is 'Find a knife, pick up the knife, find a tomato, slice the tomato, find a sink, put the knife in the sink, find the sliced tomato, pick up the sliced tomato, find a fridge, open the fridge, put the sliced tomato in the fridge, close the fridge'. There is the key frame of the planning and the full demo shows in the supplementary material.

