# OpenReview forum: "RoboGPT : An intelligent agent of making embodied long-term decisions for daily instruction tasks"
_ICLR.cc/2024/Conference — ICLR 2024 Conference Withdrawn Submission_

### Official Review · Reviewer_ygbc · 2023-10-17

**Soundness:** 2 fair
**Presentation:** 2 fair
**Contribution:** 1 poor
**Rating:** 3
**Confidence:** 4

**Summary:**

The paper proposes RoboGPT, an LLM-based approach that fine-tunes the RoboGPT planner (LLaMA 7B) for sub-goal planning,
uses RoboSkill to perform each navigation and manipulation sub-goal, and modifies the initial plan if the agent cannot find a target object.
To fine-tune the LLM, the authors follow Wang et al. to generate instructions, resulting in the dataset combined with a small amount of the generalization data.
For better segmentation mask prediction, they adopt the backbone of Fast SAM.
They outperform previous state-of-the-art methods in most metrics with noticeable margins.

**Strengths:**

- The paper tackles an important problem of ungrounded issues when utilizing LLMs in embodied agents.
- Each component of the proposed approach plays an important role as observed in the ablation study with noticeable margins.
- Exploiting commonsense knowledge encoded in LLMs for planning sounds reasonable.
- The proposed method achieves strong performance in a challenging embodied instruction following benchmark.

**Weaknesses:**

1. The proposed approach lacks novelty.
- For the RoboGPT planner, using LLMs for planning has been already explored in the same domain (Song et al., 2023) and different domains (Brohan et al., 2023a; Huang et al., 2022), as mentioned in the paper. The differences from the prior work are 1) the output format (i.e., introducing "Find, <obj>") and 2) generating more trajectories for fine-tuning but they look like engineering efforts.
- RoboSkill is quite similar to (Min et al., 2022), except for training a better segmentation module with the Fast SAM backbone, which is also an engineering effort.
- Re-plan measures the similarity between an object to be replaced with all observed objects. Though it is simple, it looks like specifically designed for the downstream task (a plan for the task is a sequence of actions and corresponding target objects). Can this approach be applied to other tasks?

2. Some comparisons are unfair or unreliable.
- The validation unseen split consists of only 50 episodes that are matched to the ground truth, which is much fewer than the original one (i.e., 821). While agreeing with the imperfections of language annotations, I do not see such a large number of imperfect annotations (i.e., more than 700 episodes). How do the authors exactly select the reduced episodes?
- While LLM-Planner uses approximately 0.5% of the original ALFRED dataset, the proposed method uses a substantial amount of datasets, about three times more than the original one. As it uses more training data, the performance gain is expected.

3. Some descriptions are not clear.
- The authors use commonsense encoded in LLMs but it is a bit counter-intuitive to augment more training data for fine-tuning, because LLMs already have much knowledge and therefore they are expected to easily adapt to downstream tasks. If we generate more data examples for planning, why not just use some vanilla planning modules (e.g., LSTM-based models)?
- For "ALFRED's data," the authors deduce 5 new tasks and construct planning templates. What are the 5 new tasks?
- What is the mapping module "S_n = Map(P_n)" here? How is it implemented?

**Questions:**

See weaknesses above.

---

> ### Author Response · Authors · 2023-11-14
>
> 1. The proposed approach lacks novelty.
>
> Answer：
> - Although some methods are currently exploring LLM for planning, we found many problems with LLM planning results through testing. There are some constraint rules that need to be obeyed when the robot performs a task, and the results of LLM planning do not satisfy the rules, especially when the task is complex. The rules include: when cutting an apple, there must be a knife in the hand; if there is something in the hand, the new object will be dropped when taking another object; the object must be turned on when putting it in the microwave or refrigerator, etc. Even if we tell the LLM about these rule constraints, the planning output by the LLM many times fails to satisfy the constraints.
> - In order to make LLM's planning more reasonable, we just collected the data of robot planning. This data was generated by ChatGPT and we did ptompt engineering to make the generated data more reasonable. The designed ptompt includes all the constraints that can be thought of for the robot, and 20 examples of typical references for robot planning. Even after ptompt engineering, the data generated by ChatGPT is not directly usable, and roughly 50\% of the data is problematic. This is why LLM-Planer (which does task planning directly with ChatGPT) doesn't score very high (16.24\% SR on Test Unseen) on the list in ALFRED, while RoboGPT agent achieved 44.57\% SR on Test Unseen.
> - Although it sounds as if producing data is an engineering thing, we also spent a lot of time and effort on Prompt design, and our Prompt is better than LLM-Planer's Prompt and makes use of self-instruct to self-generate data, eliminating a lot of manual work. Our data is the first current high-quality data for robot planning.
>
>
>
> 2. Re-plan measures the similarity between an object to be replaced with all observed objects. Though it is simple, it looks like specifically designed for the downstream task (a plan for the task is a sequence of actions and corresponding target objects). Can this approach be applied to other tasks?
>
> Answer:
> - This method is also applicable to other tasks, other areas. Our approach essentially aligns the goals in the command task with the goals in the environment based on feedback from the environment. Since the goals in the environment are obtained in real time and are not limited to a specific task or environment, it can be applied to other tasks.
>
>
> 3. Some comparisons are unfair or unreliable.
> The validation unseen split consists of only 50 episodes that are matched to the ground truth
>
> Answer：
> - To ensure fairness, we let 3 people to evaluate the correctness of the Prompter, RoboGPT, and LLM-Planer methods for each task. Doing this for all validation sets is a large task, so we chose 50 tasks. The 50 tasks were randomly selected from all validation sets, eliminating unreasonable tasks.
> 4. Some descriptions are not clear.
>
> Answer:
> - The new five types are : Pick Two \& Stack \& Place; Stack \&Cool \&Place; Pick Three\& Place; Pick Two\& Clean\& Place; Stack \&Heat \&Place
> - sub-goals $S_n=Map(P_n)$: Extract the nouns and verbs in sub-goal instructions and convert them into sub-goals.

---

> > ### Comment · Reviewer_ygbc · 2023-11-23
> > **Official Comment by Reviewer ygbc**
> >
> > I thank the authors for their response to address some concerns.
> > It addressed some of my concerns but they still remain unclear and some questions seem not answered yet.
> > Thus, I'd like to keep my original rating but may change it depending on further discussion.

---

### Official Review · Reviewer_SCK4 · 2023-10-28

**Soundness:** 3 good
**Presentation:** 3 good
**Contribution:** 2 fair
**Rating:** 6
**Confidence:** 4

**Summary:**

This paper presents RoboGPT, an agent system for solving daily instruction tasks with long-term decisions. An LLM-based planner finetuned on the proposed new robotic dataset is proposed to break a task instruction into sub-goals, and a re-plan module is designed to generate more feasible plans based on the environment feedbacks. Experiments show an improved feasibility.

**Strengths:**

1.	This paper proposes a new robotic dataset including 67k robot commands. The quality and variety of the proposed dataset is good, and experiments show reasonable improvements for models trained from it.
2.	This paper proposes a semantic map based re-plan module to solve practical issues brought by nomenclature diversity.
3.	Extensive experiments and discussions.

**Weaknesses:**

1.	The technical contribution is not clear. The improved feasibility mainly comes from 1) a self-collected and self-labelled dataset and 2) semantic map extracted by an existing work, which seems straightforward. The technical challenges and underlying designs need to be pointed out more clearly.
2.	The significance of the feasibility considered by this paper is not clear. For example, this paper proposes a re-plan module with real-time semantic recognition on failure so that the agent can know “Table” and “Desk” are the same. How about simpler solutions like a mapping between common concepts?
3.	Some works like Saycan and RT2 also consider the match of the environment and the agent ability. Key differences between the proposed method and those existing works need to be more carefully discussed.
4.	Some failure cases like “put the apple from microwave into the garbage” actually can be correctly planned by ChatGPT with proper prompts.

**Questions:**

1.	The technical contribution is not clear. The improved feasibility mainly comes from 1) a self-collected and self-labelled dataset and 2) semantic map extracted by an existing work, which seems straightforward. The technical challenges and underlying designs need to be pointed out more clearly.
2.	The significance of the feasibility considered by this paper is not clear. For example, this paper proposes a re-plan module with real-time semantic recognition on failure so that the agent can know “Table” and “Desk” are the same. How about simpler solutions like a mapping between common concepts?
3.	Some works like Saycan and RT2 also consider the match of the environment and the agent ability. Key differences between the proposed method and those existing works need to be more carefully discussed.
4.	Some failure cases like “put the apple from microwave into the garbage” actually can be correctly planned by ChatGPT with proper prompts.

---

> ### Author Response · Authors · 2023-11-14
>
> 1. The technical challenges and underlying designs need to be pointed out more clearly.
>
> Answer:
> - Although some methods are currently exploring LLM for planning, we found many problems with LLM planning results through testing. There are some constraint rules that need to be obeyed when the robot performs a task, and the results of LLM planning do not satisfy the rules, especially when the task is complex. The rules include: when cutting an apple, there must be a knife in the hand; if there is something in the hand, the new object will be dropped when taking another object; the object must be turned on when putting it in the microwave or refrigerator, etc. Even if we tell the LLM about these rule constraints, the planning output by the LLM many times fails to satisfy the constraints.
> - In order to make LLM's planning more reasonable, we just collected the data of robot planning. This data was generated by ChatGPT and we did ptompt engineering to make the generated data more reasonable. The designed ptompt includes all the constraints that can be thought of for the robot, and 20 examples of typical references for robot planning. Even after ptompt engineering, the data generated by ChatGPT is not directly usable, and roughly 50\% of the data is problematic. This is why LLM-Planer (which does task planning directly with ChatGPT) doesn't score very high (16.24\% SR on Test Unseen) on the list in ALFRED, while RoboGPT agent achieved 44.57\% SR on Test Unseen.
> -  Although it sounds as if producing data is an engineering thing, we also spent a lot of time and effort on Prompt design, and our Prompt is better than LLM-Planer's prompt and makes use of self-instruct to self-generate data, eliminating a lot of manual work. Our data is the first current high-quality data for robot planning.
> 2. The significance of the feasibility considered by this paper is not clear. For example, this paper proposes a re-plan module with real-time semantic recognition on failure. How about simpler solutions like a mapping between common concepts?
>
> Answer:
> - Mapping between common concepts should be able to do the same with noun substitution. However, our Re-plan can be re-planned for the entire goal of the plan in addition to noun substitution, e.g., the task is to take both apples and bananas to the fridge. The plan given by the planning module (take the apple and put it in the fridge, then take the banana and put it in the fridge) is not given according to the environment. Re-plan can be corrected according to the environment, e.g., if the banana is currently close to the fridge, the plan becomes to take the banana first and then take the apple.
> 3. Some works like Saycan and RT2 also consider the match between the environment and the agent ability. Key differences between the proposed method and those existing works need to be more carefully discussed.
>
> Answer:
> - The saycan is the need to call the big model at every step, wasting time and needing to know all the information about the environment. Our approach requires only one planning step for a given task and does not require knowing the environment in advance.
> - RT2 is an end-to-end approach that inputs tasks and outputs command actions. It is not easy for RT2 to migrate to new tasks and tasks with longer sequence decisions. Instead, our approach plans first for long sequence decision tasks and then invokes navigation and interaction modules to complete the task. It can be easily migrated to other arbitrary environments by simply fixing the navigation and interaction parts.
> 4. Some failure cases like “put the apple from microwave into the garbage” actually can be correctly planned by ChatGPT with proper prompts.
>
> Answer:
> - It is true that this task is relatively simple, and the current simple prompt you used gives a reasonable plan. But with the prompt you used ChatGPT would not be perfectly planned for the other examples given in the appendix. e.g., the task ’Cut a slice of bread, warm it with the microwave, put it on the counter, along with putting the knife in the cabinet’, and we give the ChatGPT: ’You are a robot given an open-ended instruction from a human, please break down the following task into individual steps that you can achieve. Instruction: ”Cut a slice of bread, warm it with the microwave, put it on the counter, along with putting the knife in the cabinet”. Skills you know are ”find”, ”pick”, ”interact with”, ”open”, ”close”, ”place”. Similar prompts work for the other cases mentioned.”  Or we give ChatGPT other prompts similar to LLM-planner's. ChatGPT still has some detail bugs that can't be directly executed by the bot. For example, when slicing bread, the knife is not held up in the hand in advance.  So, the planning of our
> complex task, which outputs planning instructions that are directly executed by the robot, is a problem that requires rigorous logic, whereas ChatGPT can only do so if the general idea is right, but there are logic errors in the details.

---

### Official Review · Reviewer_A6GH · 2023-10-30

**Soundness:** 3 good
**Presentation:** 1 poor
**Contribution:** 2 fair
**Rating:** 3
**Confidence:** 4

**Summary:**

This paper proposed RoboGPT, a large-language-model driven approach to long-horizon task planning that demonstrates state-of-the-art performance on the challenging ALFRED instruction following benchmark. RoboGPT consists of three main parts: the core RoboGPT planner (a fine-tuned Llama LLM that takes a language instruction and produces a multi-sub-goal plan), the RoboSkill skill execution module (a perception+mapping+planning module that builds a semantic map and executes plan sub-goals), and a Re-Planning module (which uses the semantic map to determine when sub-goals cannot be executed and proposed alternatives). The authors also describe the process by which they generate a 60k-instruction corpus of additional data generated in the ALFRED environment and via human effort with which they fine-tune the Llama LLM. They demonstrate state of the art performance on ALFRED and compare the performance of their approach against competitive LLM-driven baselines.

**Strengths:**

The paper proposes an interesting application of large language models to long-horizon task planning and demonstrates state of the art performance on the challenging ALFRED instruction following benchmark. Despite a reliance on significant hand-curated data and subsequent fine-tuning, there is potential for the proposed system to serve as a building block towards more reliable long-horizon planners for open-set household tasks specified in natural language.

**Weaknesses:**

**Language and Typos** It is rare that I comment specifically about language or typos in the main body of my review, yet the poor language throughout the paper makes it difficult to understand what is being proposed and how it is being evaluated. For instance, the contributions reference "the new robotic dataset", yet it is not clear if this is a dataset provided by the authors. Later on, the nature of this generated dataset and also the ablation studies are ambiguous. The abstract alone contains many typos ("palnner", "low computational", "LLMs-based") and hard-to-understand language constructions ("RoboSkill individually designed for sub-goals to learn better [...]"). The planner itself is misspelled in multiple locations: "RoboGPT agnet" and "RobOGPT". Please proofread before submitting another revision of the manuscript, as the typos and language issues made the paper difficult to follow.

**Clarifying technical contribution and differences between the proposed approach and the state of the art.** There are a few issues in the comparison with the most competitive baseline "Prompter"; the biggest potential selling point of the proposed approach over the most competitive baseline "Prompter" and so comparison to this method should not be taken lightly.

First, it is difficult to establish a clear point of comparison between the two, since they have different perception modules. The Prompter paper includes in its ablation study a version of the planner provided perfect perception, which raises performance by ~15 absolute percentage points (though it is important to note that it is only for the version of the planner that is provided low level language). The ablation study in this paper does not include a variant similar to this one in which the perception is idealized (that may be incorrect; see my questions below). Thus it is somewhat difficult to tell how much the system's performance is hampered by poor perception. Including additional results that address this omission would be incredibly helpful.

Second, one of the main claimed advantages of the proposed approach is in its ability to generalize to new prompt or task types, yet this is something that does not seem to be demonstrated. It appears that the system is trained on data the includes the new tasks, which makes it somewhat unsurprising that it is able to perform just as well on those as on the tasks from the original ALFRED dataset. By contrast, the Prompter approach cannot succeed on these, likely because its templates do not include the ability to succeed at them, yet this comparison seems unfair. How well would the RoboGPT system have done if not provided "enriched" data that includes the five additional tasks introduced by the authors? This is a key point to understand whether or not the system achieves its stated advantages.

Relatedly, it is mentioned that one of the limitations of the Prompter approach is that it is limited to a fixed number of templates and classes. However, the re-planning module in RoboGPT is also only capable of proposing terms that fall within the categories the object detection is trained to detect with the minor addition of a BERT-based similarity score to map open-set outputs to the closed-set entities on which the perception system is trained.

Finally, it seems that the performance of the proposed approach, which is fine tuned on considerable domain-specific data, is fairly similar to that of Prompter, raising questions about when one might use RoboGPT. The authors will need to justify why the proposed approach justifies the additional complexity and training needed to achieve state of the art performance.

*Deeper description of other baselines* The primary point of comparison for the proposed approach is against the LLM-Planner and the Prompter approaches. Though some discussion of these baselines is included in the Appendix, some additional discussion of how they are different, and thus why the proposed RoboGPT method should be able to succeed where they do not, would be a welcome addition to the main body of the paper.

Smaller comments and suggestions for improvement:
- The Abstract lacks any concrete metrics for success, whose addition would help clarify what the central advance of the paper is. When referencing "the proposed RoboGPT agent outperforms SOTA methods", it would be helpful to reiterate that, for example, success rate is a key metric being evaluated.
- The Appendix compares the performance of the fine-tuned perception model against what seems to be an off-the-shelf Mask RCNN. It is unsurprising that the performance of the system trained specifically in this environment would outperform one that is not and so the language in the text (In Effectiveness of RoboSkill: "the segmentation and detection of RoboSkill greatly outperform those of others") is misleading and should be removed.

**Questions:**

- The central issue that I would like the authors to clear up is the comparison to the Prompter baseline. I have detailed a couple points above that should be addressed [I will not reproduce that discussion here.]
- [reproduced from above] It seems that the performance of the proposed approach, which is fine tuned on considerable domain-specific data, is fairly similar to that of Prompter, raising questions about when one might use RoboGPT. The authors will need to justify why the proposed approach justifies the additional complexity and training needed to achieve state of the art performance.
- What is meant by the Ablation "without RobotSkill". As that module is needed to execute sub-goals, how can an ablation be performed in which this module is not provided?
- Relatedly, how well would the propose RoboGPT system perform if given access to "perfect" (ground truth) perception?
- Can the authors comment on what are the additional five task types that they added to their dataset?

---

> ### Author Response · Authors · 2023-11-14
>
> 1.  the central issue that I would like the authors to clear up is the comparison to the Prompter baseline.
>
> Answer：
> - RoboGPT agent is different from Prompter in 3 ways：
>   -  It adds RoboGPT's plan module, which allows you to plan almost all of your family's daily task commands. Although it uses more data, the plan model can also handle other categories of richer household tasks and tasks from other domains (not seen in the training set), such as 'how to make bread or how to write an essay'.
>   -  The perception module is different from the Prompter in that we re-trained the perception module and compared it to the Prompter's original Mask RCNN (trained in ALFRED data), so the segmentation model comparisons in the appendix are fair comparisons as all the models were trained in the training set.
>   - the replan module, which maps objects in the task description to objects in the environment, solving the problem of open vocabulary, e.g. 'sidetable' and 'shelf'.
>
> 2. The authors will need to justify why the proposed approach justifies the additional complexity and training needed to achieve state of the art performance.
>
> Answer：
> - Prompter is only capable of handling 7 types of tasks, and our RoboGPT can handle many more daily tasks. In real life, there can be thousands of types of daily tasks, so planning models that can handle more and more complex tasks are needed.
> - For the 7 types of tasks in ALFRED, Prompter used all the data in the training set to train the Bert model to classify the types of tasks and the target containers used etc. Prompter then matched the classification results with the 7 templates to get the planned results. Therefore, Prompter is only able to handle 7 types of daily tasks.
> - In real life, there can be thousands of types of daily tasks, and it would be a waste of time to design so many templates. And we need to collect enough data to train Bert in Prompter to classify the tasks. The more types of tasks there are, the more difficult it is to classify them, and the performance of Prompter's templates is difficult to guarantee.
> - Our approach uses 6.7k of self-instruction data, 12 classes of data constructed in ALFRED, and publicly available data easily accessible online to fine-tune the LLama model. Our trained LLama can then quickly learn more types of data.
>
> 3. What is meant by the Ablation "without RobotSkill". As that module is needed to execute sub-goals, how can an ablation be performed in which this module is not provided?
>
> Answer:
> - Sorry for such an inaccurate representation. without RobotSkill means to use Prompter's perception and interaction module, where the main difference is the segmentation module and the map building part. RoboSkill uses Fast-SAM's segmentation  instead of MaskRcnn, which improves  perception performance. RoboSkill builds the map taking into account replan, embodying all objects related to the goal in the map, in order to allow the agent to replan based on the seen objects. The prompter, on the other hand, only embodies the objects that appear in the goal, which is not conducive to a clear idea of what all objects in the scene are currently seen by the robot.
>
>
>
> 4. Relatedly, how well would the propose RoboGPT system perform if given access to "perfect" (ground truth) perception?
>
> Answer:
>  - We do experiments and the results of the method with segmentation ground truth achieve 5\% improvements and the results of the method with  depth and segmentation ground truth achieve 12\% improvements.
>
> 5. Can the authors comment on what are the additional five task types that they added to their dataset?
>
> Answer:
> - The new five types are: Pick Two \& Stack \& Place; Stack \&Cool \&Place; Pick Three\& Place; Pick Two\& Clean\& Place; Stack \&Heat \&Place

---

### Official Review · Reviewer_16gg · 2023-11-01

**Soundness:** 2 fair
**Presentation:** 1 poor
**Contribution:** 3 good
**Rating:** 3
**Confidence:** 4

**Summary:**

This work has three main contributions: 1) a synthetic robot planning dataset with generated instructions and sub-goal samples that are augmented from the ALFRED starting dataset, 2) RoboGPT, a LLM-based high level planner that finetunes on the synthetic dataset in 1) and produces subgoal text instruction plans given a high level text instruction; RoboGPT is claimed to surpass ChatGPT and other methods, and 3) claimed SOTA performance of 2) on the ALFRED benchmark. The synthetic dataset starts with 360 sampled ALFRED plans and uses them as few-shot prompts for an LLM to generate new instruction-plan pairs; in total, there are 7,724 samples generated with this method. The RobotGPT planning agent trains on the 7k synthetic samples in addition to 60k samples from ALFRED. To utilize RoboGPT in the ALFRED benchmark, this work introduces two modules: a RoboSkill module that utilizes perception modules to generate voxel and segmentation masks for querying robot actions, and a Re-Plan module that utilizes VLM object detections to update low-level robot primitives to re-map object names. They demonstrate this combined system on the ALFRED benchmark, and compare against other prompting and planning works.

**Strengths:**

- The method achieves strong performance on ALFRED
- The new instruction dataset combining ALFRED data with synthetic data is a good contribution for robot planning methods that need to produce language primitive subgoals given a high-level text instruction

**Weaknesses:**

- The core motivating limitations of current zero-shot LLMs at robot planning are not justified. The paper claims that RoboGPT is the first model to understand prefixes, object dependencies, and object quantities; however, simple tests of SOTA LLMs (which are not finetuned on robot data) are able to easily solve these tasks.  I tried to ask ChatGPT as well as Bard to respond to the examples provided in the paper and it works on the very first prompt I use: `You are a robot given an open-ended instruction from a human, please break down the following task into individual steps that you can achieve. Instruction: "There is a stove and no microwave, how to heat an apple". Skills you know are "find", "pick", "interact with", "open", "close", "place"`. Similar prompts work for the other cases mentioned. I did no prompt engineering for this. This is highly worrying, and goes against the claim "RoboGPT with strong generalization ... surpass[es] ChatGPT and other planning methods". Therefore, this results in two serious issues: 1) the main motivating claim of why RoboGPT is needed is not supported, 2) if anyone today can verify independently that zero-shot LLMs can perform well on the exact examples included in the paper, then the results in Table 1 and Table 2 seem quite dubious.
- The large performance gains in ALFRED may be attributed to the domain-specific Map() module in Section 3.1 or the visually grounded feedback in Section 3.2; however, from the main contributions, a core claim is the RoboGPT model is better than baselines on high-level instruction to low-level subgoal instruciton prediction. However, it seems that Re-Plan is actually providing the bulk of the gains. In fact, the ablation in Table 1 of "RoboGPT without Re-Plan" performs worse than the baselines on some benchmarks, despite "RoboGPT without Re-Plan" containing supposedly superior subgoal instruction prediction.
- The presentation is extremely poor, with a high volume of grammatical and spelling errors. See below for some examples. In addition, the paper is convoluted and quite difficult to follow, with RoboSkill and Re-Plan modules seemingly very separate from the finetuned LLM in RoboGPT, and yet being quite domain-specific modules themselves, yet with few details. In general, the level of polish in the manuscript is extremely far from the level expected in an ICLR publication.
- There is no explanation of the text to low-level action execution in RoboSkill.
- The core claim is not clear. Is RoboGPT the finetuned LLM model (based on Introduction)? Or does RoboGPT also include RoboSkill and Map and Re-Plan (based on Table 1)? If RoboGPT includes RoboSkill and Re-Plan, then it needs to be compared/discussed against works that also re-plan (see below).
- There is no comparison against works that re-plan in robotics, including those that take visual scene information leveraging perception modules. For example, Inner Monologue [1] incorporates re-planning in the form of text-based VLM observations.
- Minor writing issues (an automated spellchecker may catch additional issues):
    - Abstract: "with re-plan" => "with re-planning", "palnner" => "planner" x2
    - Introduction: "detialed" => "detailed", "object name match" => "object name matching", "as far as we known" => "as far as we know"
    - 3.3 RoboSkill: "and interaction module)" => "and interaction module"
    - 4 Experiments: "LLM-Planer" => "LLM-Planner"
    - 4.2 Ablations: "RoboGOPT" => "RoboGPT"


[1] "Inner Monologue: Embodied Reasoning through Planning with Language Models", Huang et al. 2022

**Questions:**

- Clarifications to my questions above will be appreciated and help me better understand the work.

---

> ### Author Response · Authors · 2023-11-14
>
> 1. Questions about the ChatGPT being able to complete "There is a stove and no microwave, how to heat an apple" and the dubious results of Table 1 and Table 2.
>
> Answer:
> - It is true that this task is relatively simple, and the current simple prompt you used gives a reasonable plan. But with the prompt you used ChatGPT would not be perfectly planned for the other examples given in the appendix and ALFRED tasks.  This is why LLM-Planner (which does task planning directly with ChatGPT) doesn’t score very high (16.24\% SR on Test Unseen) on the list in ALFRED, while RoboGPT agent achieved 44.57\% SR on Test Unseen. There are some constraint rules that need to be obeyed when the robot performs a task, and the results of LLM planning do not satisfy the rules, especially when the task is complex. The rules include: when cutting an apple, there must be a knife in the hand; if there is something in the hand, the new object will be dropped when taking another object; the object must be turned on when putting it in the microwave or refrigerator, etc. Even if we tell the LLM about these rule constraints, the planning output by the LLM many times fails to satisfy the constraints.
>
>
>   - For example: the task 'Cut a slice of bread, warm it with the microwave, put it on the counter along with putting the knife in the cabinet', and we give the ChatGPT: 'You are a robot given an open-ended instruction from a human, please break down the following task into individual steps that you can achieve. Instruction: "Cut a slice of bread, warm it with the microwave, put it on the counter along with putting the knife in the cabinet". Skills you know are "find", "pick", "interact with", "open", "close", "place". Similar prompts work for the other cases mentioned.'
>   - The output of ChatGPT will vary a bit each time, we tested it 10 times and chose the best one：
>   - ChatGPT plans: '1、Find a Knife; 2、Pick Up the Knife; 3、Open the Cabinet; 4、Place the Knife in the Cabinet; 5、Find a Loaf of Bread;6、Find a Cutting Board; 7、Place the Cutting Board on the Counter; 8、Cut a Slice of Bread; 9、Find a Microwave-Safe Plate; 10、Place the Bread on the Plate; 11、Warm the Bread in the Microwave; 12、Find Oven Mitts or a Towel; 13、Place the Warm Bread on the Counter; 14、Close the Microwave; 15、Close the Cabinet'
>
>    - Steps 1-4 of the planning given by ChatGPT are to put the knife in the cupboard, whereas what we want is to use the knife to cut the bread. Steps 5-8 of the plan is to cut the bread, and there is no detailed operation that the robot can perform, but after finding the bread, it cuts it directly, and does not use the utility knife to cut the bread.Step 11 is to heat up the bread directly, and there is no operation to turn on the microwave and put the bread in.
>    - The planning of our model RoboGPT is: first find the knife and pick up the knife, find the bread and cut the bread, then find the microwave oven, open the microwave oven and put it in.The planning of our model is logical for robot operation and can be used directly by the robot.
>
> - The results in Table 2 are taken directly from below the ALFRED list. Our method is the highest of all methods with papers. The results of Table1 also matches the results described in the LLM-planner paper , which uses ChatGPT to plan tasks.
>
> 2. Question about the large performance gains in ALFRED using RoboGPT.
>
> Answer:
> - The RoboGPT planning module is meant to generalise to other categories of task planning, not just limited to the 7 categories of objectives in ALFRED. As can be seen in last column of Table 1, RoboGPT has an accurate success of planning on Gen. Task.
> RoboSkill is meant to have better perception to improve the success rate of subtask execution.
> - The Replan module is to make the subtasks more contextual. The robot can find the desired goal faster. So comparing RoboGPT agent without Replan its RLWSR metrics are a little bit lower than RoboGPT with replan, but still higher than the baseline prompter.
> - RoboSkill is meant to have better perception to improve the success rate of subtask execution.
> - The Replan module is to make the subtasks more contextual. The robot can find the wanted target faster. So RoboGPT agent without Replan is a little bit lower in RLWSR metrics than RoboGPT with replan and the baseline Prompter.
> 3. The presentation is poor, with a high volume of grammatical and spelling errors.
>
> Answer： Thank you for your feedback. We acknowledge the issues raised and will thoroughly revise the paper to address the poor presentation, grammatical errors, and spelling mistakes.

---

> > ### Comment · Reviewer_16gg · 2023-11-22
> >
> > Thanks for your response.  You have addressed some of my questions but my main concerns still remain.
> >
> > I keep my rating for now but may change it during discussion with other reviewers.